# Generating Compositional Scenes via Text-to-image RGBA Instance Generation

**Alessandro Fontanella**[*]
University of Edinburgh

**Petru-Daniel Tudosiu**
Huawei Noah's Ark Lab

**Yongxin Yang**
Huawei Noah's Ark Lab

**Shifeng Zhang**
Huawei Noah's Ark Lab

**Sarah Parisot**[*]
Microsoft Research

## Abstract

Text-to-image diffusion generative models can generate high quality images at the cost of tedious prompt engineering. Controllability can be improved by introducing layout conditioning, however existing methods lack layout editing ability and fine-grained control over object attributes. The concept of multi-layer generation holds great potential to address these limitations, however generating image instances concurrently to scene composition limits control over fine-grained object attributes, relative positioning in 3D space and scene manipulation abilities. In this work, we propose a novel multi-stage generation paradigm that is designed for fine-grained control, flexibility and interactivity. To ensure control over instance attributes, we devise a novel training paradigm to adapt a diffusion model to generate isolated scene components as RGBA images with transparency information. To build complex images, we employ these pre-generated instances and introduce a multi-layer composite generation process that smoothly assembles components in realistic scenes. Our experiments show that our RGBA diffusion model is capable of generating diverse and high quality instances with precise control over object attributes. Through multi-layer composition, we demonstrate that our approach allows to build and manipulate images from highly complex prompts with fine-grained control over object appearance and location, granting a higher degree of control than competing methods.

## 1 Introduction

The development of text-to-image generative diffusion models has allowed the generation of high-quality synthetic images guided by textual prompts. However, achieving desired image quality and properties inherently requires tedious and meticulous prompt engineering [35]. As image content is solely described using a textual description, prompt content has to be carefully crafted, notably taking into account the models' struggles to understand subtleties of language (e.g. counting, object attributes, negation or spatial relationships) [16]. Additionally, minor prompt modifications can lead to substantial modifications of generated image content, further increasing prompt crafting tediousness.

An intuitive way of improving controllability of generated content is by providing alternative image descriptors, e.g. using image layouts which describe the location of specific objects in an image as bounding boxes. Several approaches have been proposed to inject layout information in the generative process. Training based methods [24, 54] adapt or introduce new model weights to introduce object coordinates as conditioning, while [8, 51] leverage cross-attention manipulation

---

[*]Work done while at Huawei Noah's Ark Lab

38th Conference on Neural Information Processing Systems (NeurIPS 2024).

and inference time optimisation. While substantially increasing control over image structure, these solutions still generate all components as a single denoised image. This limits the user's ability to finely control object attributes, and layout adjustments requires re-generation with limited content preservation guarantees. These issues can potentially be addressed with image editing techniques. This, however, can require expensive data construction and training [4], while the displacement or resizing of objects (essential for layout manipulation) remains a very challenging type of edit that few have explored [9, 31]. This challenge can be attributed to the fact that objects and backgrounds have to be regenerated, while at the same time ensuring appearance preservation with the original image.

The emerging layer-based generation paradigm [38, 49, 40] has shown promise in addressing these limitations. The idea is to represent images as multi-layer stacks, where different image components are generated on separate layers. This has two key benefits: 1) instances can be generated more precisely as separate images with individual prompts, 2) separating image components simplifies image manipulation tasks, as only the relevant layers will be modified. Nonetheless, layer-based methods typically suffer from the fact that layers are not fully generated, as they are collapsed to a single image in the later stages of a diffusion process. Generating all components jointly can affect the ability to achieve fine-grained control of layers attributes, control their relative positioning in 3D space (as layers are collapse with no hierarchical structure), ensure a smooth composition of all image component, and achieve layout manipulation abilities with strong content preservation (as layers and composite image influence each other).

In this work, we seek to address current limitations through a multi-stage generation process that is designed for fine-grained control, flexibility and interactivity. We propose to build complex scenes by first generating individual instances as RGBA images, then iteratively integrating them in a multi-layer composite image according to a specific layout. The ability to generate individual instances allows to finely control instance appearance and attributes (e.g. colours, patterns and pose), while their layered composition allows to easily control and modify positioning, scale and ordering. In particular, our two-stage approach provides intrinsic layout and attribute manipulation abilities with strong content preservation. To achieve our goal, we firstly train a diffusion model capable of generating RGBA images, i.e. images with alpha transparency information. Directly generating isolated instances, c.f. generating and extracting them with a segmentation model, ensures better transparency masks and more fine-grained control of instance attributes. Our RGBA generator is obtained by fine-tuning a latent diffusion model (LDM) using RGBA instance data from the recently released MuLAn dataset [47]. In order to incorporate transparency information in the generative process, we devise a transparency-aware training procedure for both VAE and diffusion model. In contrast with the contemporary transparent generation method of [57], which implicitly encodes transparency information in the null space of the VAE to carry out standard LDM fine-tuning, we explicitly integrate transparency in our generation and training process. Our VAE is trained so as to disentangle RGB and alpha channels, ensuring colour and detail preservation in RGB reconstruction. Our LDM is then fine-tuned with a novel training paradigm leveraging our disentangled latent space, that allows a conditional sampling-driven inference where alpha and RGB latents are sequentially denoised with mutual conditioning. Finally, we leverage our RGBA generator to build composite images with fine-grained control over object attributes and scene layout. We build multi-instance scenes via a multi-layer noise blending solution, where each instance is associated to a specific image layer. Crucially, each instance is integrated in the scene one layer at a time, yielding increasingly complex image layers so as to ensure scene coherence and accurate relative positioning. This contrasts with pre-existing multi-layer works which assemble all instances at once [38, 49], and is uniquely afforded by our RGBA generation process. By manipulating latent representations in early stages of the denoising process, we are able to achieve high degrees of precision and control, while at the same time generating smooth and realistic scenes. An overview of our complete methodological process is shown in Fig. 1.

We provide a thorough evaluation of our RGBA generator's capabilities, showing that we are able to generate highly diverse objects and precisely control their attributes and style, outperforming alternative solutions. In addition, our scene composition experiments show that we consistently outperform state of the art layout controlled methods, while highlighting our unique ability to generate and manipulate complex scenes with strongly overlapping objects. In summary, our main contributions are the following:

- We propose a multi-layer generation framework designed for fine-grained controllable generation that grants user control over instance appearance and location, with intrinsic scene manipulation capabilities.

- We introduce a novel fine-tuning paradigm for pre-trained diffusion models to generate individual objects as RGBA images with transparency information. We propose a disentangled training strategy that relies on mutual conditioning of RGB and alpha channel latents.

- We develop a multi-layer scene compositing strategy based on noise blending that allows to build and manipulate complex scenes from multiple instance images. By combining RGBA generation with scene compositing, we are able to generate images with accurate attribute assignment and relative object positions.

## 2  Related Work

**Text-to-image and transparent generation.** GLIDE [32] first proposed text-to-image generation with diffusion models by adopting classifier-free guidance through text. While GLIDE trains the text encoder jointly with the diffusion prior using paired image and text data, Imagen [43] employs a frozen large language model as the text encoder. More recently, Latent Diffusion Models (LDM) trained in latent space have increased in popularity, given their lower computational requirements[41]. Stable Diffusion incorporates an adversarial objective to learn the latent representation and enhance the authenticity of generated images and introduces cross-attention for text conditioning. While the U-Net [42] architecture was originally the most popular for latent diffusion models, recent works are converging towards the use of Diffusion Transformers (DiT) [36], which operate on latent patches. For example, Pixart-$\alpha$ [6] has demonstrated remarkable image generation capabilities, while reducing the computational requirements to $10.8\%$ of Stable Diffusion-v1.5's training time. Leveraging these models to generate instances with transparency information has been rarely explored. Text2Layer [60] generates foreground and background images separately, defining foreground as the ensemble of all salient objects and predicting the foreground alpha mask separately from RGB pixels. Developed concurrently, LayerDiff [18] generates image components as separate images, providing the transparency masks as input. This technique tends to generate partial instances, as occluded areas are not generated. Also contemporary, LayerDiffusion generates transparent images by encoding transparency in the null space of the pre-trained VAE, allowing to fine-tune a diffusion model using standard techniques. While able to generate high quality instances, this implicit transparent modelling can lead to inconsistent results, without guarantees that transparency will be achieved. In contrast, our explicit disentangled latent space allows for more control and guarantees transparent images outputs.

**Image editing and scene controllability** The increasing popularity of diffusion models has highlighted the limitations of working solely with text based control. We identify two main areas of research seeking to increase generative controllability: image editing and layout controlled generation. Image editing can be achieved by fine-tuning on dedicated dataset [4], constrained sampling with CLIP [39] losses [20, 21, 45, 17, 50], mask guided inpainting [30, 59, 11, 52, 7], training-free cross attention manipulation [12, 34, 5, 48], and inference-time optimisation [9]. Despite achieving impressive results for replacement and local modification tasks, image editing often struggles with more complex manipulations such as moving, rescaling and removing, especially on non isolated instances. By modifying pre-generated content, editing methods additionally have limited layout control capabilities. Alternatively, precise layout control during generation has been explored by introducing additional conditioning. GLIGEN [24], ReCo [54] and Boxdiff [51] achieve image generation from layouts given as bounding boxes specified by the user. ControlNet [58] allows to incorporate several forms of image based conditioning, such as sketches, depth maps, or canny edges. These techniques afford high generative controllability, but lack layout editing capability. Composite [2] and layer-based generation techniques [38, 49, 40] offer a promising avenue, where individual scene components are associated with unique prompts and assembled in a scene with bounding boxes or instance specific masks, leveraging the diffusion model's intrinsic priors to build coherent scenes. However, generating scenes and all individual components jointly requires compromising between instance quality, attribute accuracy and scene coherence, reducing flexibility and control. We build from layered approaches, and use multi-layer paradigms to iteratively integrate pre-generated RGBA instances in increasingly complex images, affording us precise control over layout, instance attributes and allowing us to focus on scene composition quality solely during the composition step.

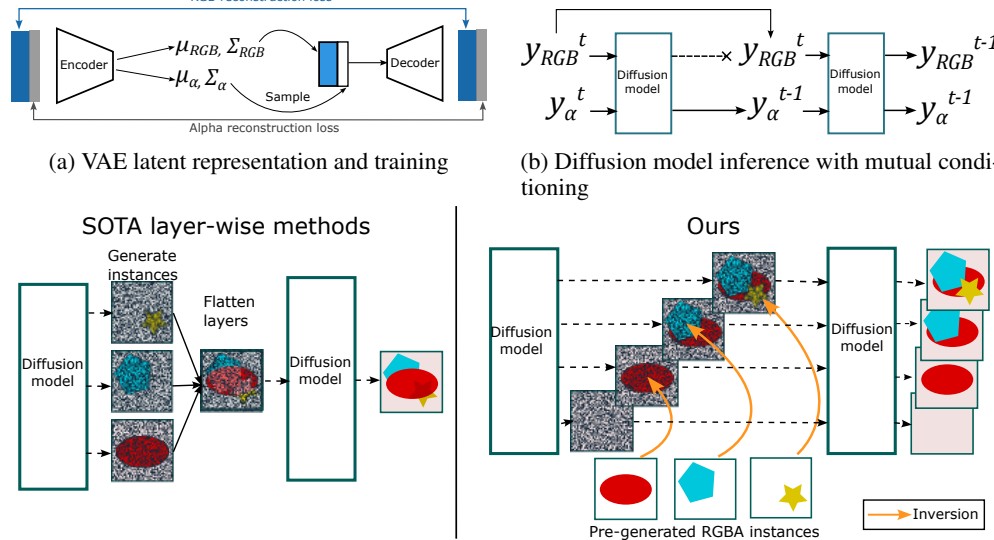

(a) VAE latent representation and training

(b) Diffusion model inference with mutual conditioning

(c) Our scene composition process compared to state of the art multi-layer methods

Figure 1: Overview of key components of our proposed methodology.

# 3 Methods

## 3.1 Preliminaries

**Diffusion models.** Diffusion models learn to reverse a *forward diffusion process*, where images are iteratively converted to near isotropic Gaussian noise over $T$ timesteps as $y_t = \sqrt{(1 - \beta_t)}y_{t-1} + \beta_t \epsilon$, where $\epsilon \sim \mathcal{N}(0, I)$ and $\beta$ describes a noise schedule. Diffusion models are trained to reverse the process by learning to predict the noise $\epsilon_\theta$ that was added at a specific timestep:

$$\mathcal{L} = \mathbb{E}_{y,\epsilon,t} = \|\epsilon - \epsilon_\theta(y_t, t, C)\| \tag{1}$$

Where $C$ is a conditioning variable, typically a text prompt embedding. Starting from isotropic Gaussian noise $y_T$ at inference time, diffusion models generate images through the reverse iterative denoising process, where $y_0$ is the final denoised output. At a given timestep $t$ of this *backwards diffusion process*, the pre-trained model predicts the noise $\epsilon_\theta$ that updates current noisy image $y_t$ to move on to the next timestep $t - 1$. In this work, we consider *latent diffusion models* (LDMs), where the diffusion process is carried out in the latent space of a Variational AutoEncoder (VAE) [41].

Alternatively, the pre-trained model can be leveraged to convert an image $x_0$ to Gaussian noise by iteratively adding predicted noise $\epsilon_\theta(x_t, t)$ according to the same noise schedule [44]. This *inversion* process provides an estimation of the input noise $x_T$ that allows to obtain $x_0$ via the backwards diffusion process. This technique is particularly relevant to image editing tasks and noise blending methods like ours. To differentiate between both processes, we use $y_t$ to refer to *generated* latents and $x_t$ to refer to *inverted* latents in the remainder of this paper.

**RGBA representation.** An RGBA instance image is modelled as a four channel image $I \in \mathcal{R}^{w,h,4}$, where the additional *Alpha* channel describes the level of transparency of RGB pixels. The alpha channel takes values in $[0, 1]$, where 0 correspond to full transparency. Transparent pixels for which we do not have RGB information are set to black ($RGB = (0, 0, 0)$).

## 3.2 RGBA Instance Generation

A simple method for creating instances with transparency involves using a text-to-image model to generate an initial image, then applying image matting techniques [55] to extract the transparency alpha mask. This reliance on automated matting can yield inaccurate masks, particularly when dealing with complex object structures and image backgrounds. In addition, this approach reduces controllability, as object attributes can bleed in the background content rather than the instance itself

[16]. Alternatively, we propose to natively generate RGBA instances using a generative diffusion model. To this end, we developed a transparency aware training procedure for a pre-trained LDM that focuses on the interaction between RGB and alpha channels. This section will first introduce our training dataset, then detail our fine-tuning approach for the LDM's VAE and diffusion model.

**Training data.** We employed 87989 instances from the MuLAn dataset [47], a novel dataset consisting of automatically generated RGBA decompositions with a diverse array of scenes, styles and object categories, and 15791 instances extracted from a variety of image matting datasets with high quality masks. More details about the datasets are provided in Appendix A.1.

**RGBA VAE.** We train our RGBA VAE following the training procedure of [41], comprising a $L_1$ reconstruction loss $\mathcal{L}_{recon}(I, \mathcal{D}(\mathcal{E}(I)))$, a perceptual loss $\mathcal{L}_{LPIPS}(I, \mathcal{D}(\mathcal{E}(I)))$, a discriminator based adversarial loss $\mathcal{L}_{adv}(\mathcal{D}(\mathcal{E}(I)))$, and a Kullback Liebler (KL) loss between the estimated latent distribution and a normal distribution $\mathcal{N}(x : 0, 1)$. However, we make key modelling and training adjustments in order to accommodate for the new alpha transparency information.

Firstly, the additional channel is simply taken into account by replacing and retraining the input and output layers of the model. Secondly, we observed that learning a joint RGBA latent space leads to entanglement of RGB and alpha channels, affecting generation capability of diffusion models trained in this latent space. We address this challenge by disentangling representations in the latent space: our VAE predicts two separate distributions $\mathcal{N}(x : \mu_{RGB}, \Sigma_{RGB})$ and $\mathcal{N}(x : \mu_\alpha, \Sigma_\alpha)$, each associated with a separate KL loss. While we do not explicitly model the separate distributions to encode RGB and alpha channels respectively, it is a natural disentanglement of the data.

Lastly, the VAE in [41] is trained with very small KL regularisation (e.g. a weight factor of $w_{KL} = 10^{-6}$ on the KL loss), so as to focus on reconstruction quality. In our setting, we observed that this deviation from standard VAE training was harmful to generative ability, yielding dark and highly contrasted images. In contrast, training our VAE with large regularisation ($w_{KL} = 1$) noticeably improved image quality, while at the same time enforcing disentanglement more strongly (see Appendix A.3 for visual comparisons). The perceptual loss is also computed separately for RGB and alpha channels and the two components $\mathcal{L}_{LPIPS}(I_{RGB}, \mathcal{D}(\mathcal{E}(I))_{RGB})$ and $\mathcal{L}_{LPIPS}(I_\alpha, \mathcal{D}(\mathcal{E}(I))_\alpha)$ are then averaged.

**RGBA Diffusion Model.** In the second stage, we fine-tune the LDM on our instances datasets. Our VAE fine-tuning keeps the dimension of the original LDM latent space (4 channels and $64 \times 64$ spatial dimension), allowing us to directly fine-tune without architecture adaptation. However, in our case, the latent space also encodes information on the transparency layer. When sampling from our model, we seek to exploit the mutual dependency between RGB and alpha channels. In particular, given noised latents $y_t^{RGB}$ and $y_t^\alpha$ for RGB and alpha channels respectively at timestep $t$, information contained in $y_{t-1}^{RGB}$ could be employed to inform the update from $y_t^\alpha$ to $y_{t-1}^\alpha$ and vice-versa. In order to train the network to leverage this conditional information, we modify the training procedure of the LDM. Given $y_0^{RGB}$ and $y_0^\alpha$, we sample two different Gaussian noise vectors $\epsilon^{RGB}$ and $\epsilon^\alpha$ and use them to compute $y_t^{RGB}, y_{t-1}^{RGB}, y_t^\alpha, y_{t-1}^\alpha$ with the forward process of the diffusion model and $t$ randomly sampled in $[0, T]$, with $T$ number of training steps. Our network is trained to predict $(\epsilon^{RGB}, \epsilon^\alpha)$ jointly given one of the following as input: $(y_t^{RGB}, y_t^\alpha)$, $(y_t^{RGB}, y_{t-1}^\alpha)$, or $(y_{t-1}^{RGB}, y_t^\alpha)$.

This training procedure unlocks the ability to perform conditional sampling. Given $(y_t^{RGB}, y_t^\alpha)$, we can alternate between updating the alpha component $y_{t-1}^\alpha$ and then use $(y_t^{RGB}, y_{t-1}^\alpha)$ to update the RGB component, obtaining $(y_{t-1}^{RGB}, y_{t-1}^\alpha)$. Alternatively, we can update the RGB component first and use it to condition the alpha update, but observed that the former approach worked best in practice. When sampling from diffusion models, it is common to use fewer sampling steps than at training time for faster image generation [53]. Therefore, in order to make our training regime more flexible and applicable to a variety of sampling strategies, we additionally use pairs $(y_t^{RGB}, y_k^\alpha)$, and $(y_k^{RGB}, y_t^\alpha)$ as conditioning input in the second half of training iterations, with $k$ randomly sampled in $[0, t-1]$.

### 3.3 Multi-Layer Noise Blending for Scene Composition

We consider that we have an image layout available, and generated $K$ instances $\{I_k, M_k\}$ using our RGBA generator, where $I_k$ refers to RGB values and $M_k$ is the transparency alpha mask. We represent a layout as a collection of bounding boxes, where each box is associated with a specific instance: $\mathcal{L} = \{[cx_k, cy_k, w_k, h_k], \text{for } k \in 1, \cdots, K\}$. We design our scene composition approach

as a multi-layer noise blending process, where instances are sequentially integrated in intermediate layered representations. As a result, we concurrently generate $K + 1$ images (sorted as background, and $K$ composite images with increasing number of instances). While more costly, we observed that generating layered images affords more flexibility, better control over relative positions of instances and yields more natural compositions. An algorithmic overview is provided in Appendix B.

We consider the latent representation $x_0^k$ of instance image $I_k$, and compute its noisy representation $x_t^k$ at all diffusion timesteps $t \in [1, T]$ using DDIM inversion [44]. To generate our layered composite image, we first initialise our $K + 1$ images with the same noise vector $y_T$. Inspired from compositing [2] and layered [38] image generation methods, we combine noisy image representations for the first $n$ timesteps of the diffusion denoising process. After carrying out the denoising update at time $t$, and before moving on to the next time step, we sequentially update noisy images as follows:

$$y_t^k = y_t^{k-1} \cdot (1 - m_k) + x_t^k \cdot m_k \text{ for } k \in [1, K] \tag{2}$$

where $m_k$ is the instance alpha mask $M_k$ downsampled to the latent space and $y_t^0$ is the background image generated concurrently. Equation 2 builds layered images by iteratively injecting new instances in the noisy images at the desired locations. Carrying this process only at the first $n$ generative timesteps enables to precisely control scene composition and instance appearance, while at the same time allowing for a smooth and realistic composition. The smaller $n$ is, the more freedom is given to the generative model to adapt scene content. We further detail two optional additional mechanisms.

**Background blending.** In the noise blending procedure described above, the background is generated independently from the rest of the image. That can reduce blending quality as generated background elements can be incompatible with instance locations. We propose to address this limitation by introducing background blending. We inject appearance information from the composite image to the background as:

$$y_t^0 = y_t^0 \cdot m^* + \frac{1}{2}(y_t^0 + y_t^K) \cdot (1 - m^*) \tag{3}$$

where $m^*$ is the union of all instances alpha masks. The background area of the composite image gets adjusted throughout the generation timesteps to accommodate instances neighbourhood. Equation 3 has two benefits: 1) it ensures stronger consistency between the background layer and final composite image, and 2) allows the background image to be generated with stronger awareness of instances. This blending process can be carried out for the first $b$ steps of the composition process.

**Increasing cross-layer consistency.** In situation where intermediate layers are needed, and $n$ is small, consistency can be enforced on the composite image by applying Eq. 2 to $y^K$ for $n_s$ subsequent timesteps: $y_t^K = y_t^K \cdot (1 - m_k) + y_t^{k-1} \cdot m_k$ for $k \in [1, K - 1]$.

**Scene editing.** Our design intrinsically allows scene manipulation and editing easily. Once instances are pre-generated, scene content can easily altered locally by replacing instances, or modifying instance location by providing new bounding boxes. The modified image can easily be generated following our scene composition process with a fixed seed, without any additional constraints.

## 4 Experiments

### 4.1 RGBA generation

Our RGBA generator is fine-tuned from a pre-trained PixArt-$\alpha$ model [6]. As baselines, we compare quantitatively and qualitatively to Stable Diffusion 1.5 (SD-1.5) and PixArt-$\alpha$, adding 'on a black background' to the captions to replicate instance generation. We additionally compare to both models combined with the Matte Anything (MA) matting algorithm [55], our reimplementation of Text2Layer [60], and LayerDiffusion [57]. In all approaches we used 100 steps. Our RGBA generator leverages the conditional sampling approach described in Sec. 3.2.

In Fig. 2 and 3 we show qualitative examples of instances obtained with our RGBA generator. Fig. 2 shows that we are able to generate instances across different styles and to follow fine-grained attributes in prompts. Fig. 3 provides visual comparisons to our transparent baselines. We can observe that our approach is able to generate realistic instances following the instructions given. Text2Layer shows lower image quality and excessive transparency, while LayerDiffusion struggles to follow prompt details, such as image style. Combining SD with Matting allows to achieve reasonable segmentation of the instances generated, while when applied to PixArt-$\alpha$ it can sometimes struggle

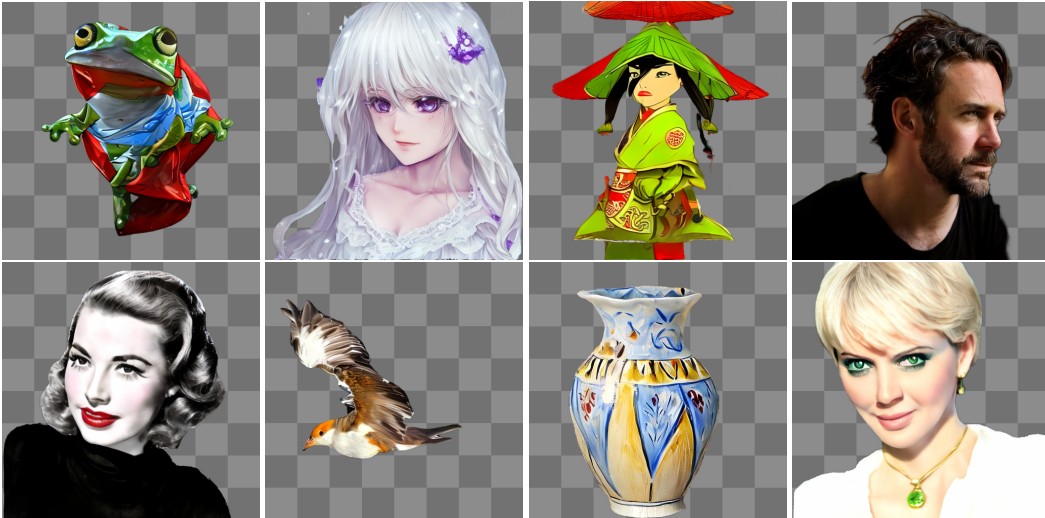

Figure 2: Our model can generalise to different styles and to follow detailed instructions. Top row: 'a **cartoon style** frog', 'a digital artwork of an **anime-style** character with **long, flowing white hair** and **large and expressive purple eyes** in a **white attire**', 'a **stylised** character with a traditional Asian **hat, with a red and green pattern**', 'a man with a **contemplative expression** and a **neatly trimmed beard**', Bottom row: 'a woman with a **classic, vintage style, curly hair, red lipstick**, fair skin in a dark attire', 'a bird **mid-flight** with **brown and white feathers** and **orange** head', 'a **hand-painted ceramic vase** in **blue and yellow colours** and with a **floral pattern**', 'a woman with **short, blonde hair**, vivid **green eyes**, in a **white blouse**, with a **gold necklace** featuring a pendant with a **gemstone**'.

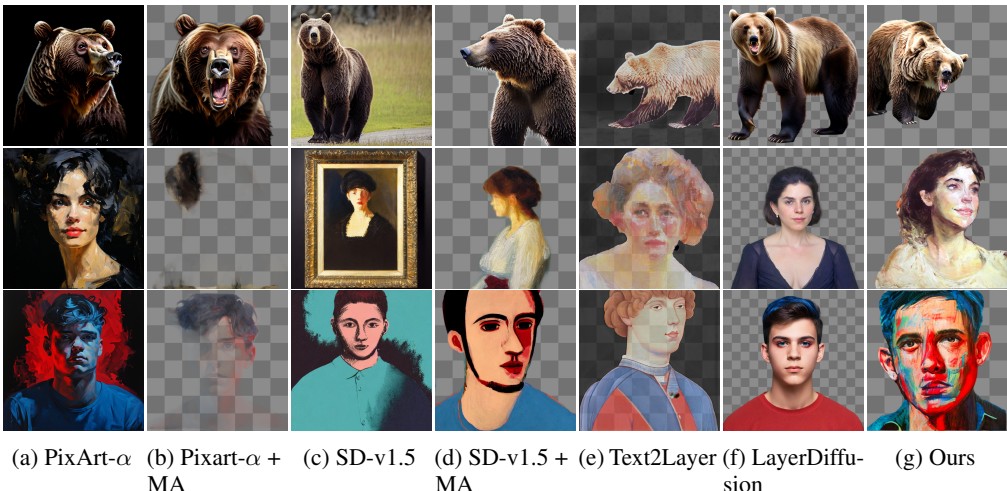

(a) PixArt-$\alpha$    (b) Pixart-$\alpha$ + MA    (c) SD-v1.5    (d) SD-v1.5 + MA    (e) Text2Layer    (f) LayerDiffusion    (g) Ours

Figure 3: Instances generated with the captions: 'a majestic brown bear with dark brown fur, its **head slightly tilted** to the left and its **mouth slightly open**', 'an **Impressionist** portrait of a woman', 'a portrait of a young man, depicted in a **blend of blue and red tones**'.

to correctly identify and segment the main object of the image, especially when dealing with artwork content. On top of attributes bleeding in the background, this highlights how unreliable matting can be for instance generation purposes.

We quantitatively evaluate the quality of the generated instances with the Kernel Inception Distance (KID) [3], a metric that was proposed to reduce the bias of the Fréchet Inception Distance, especially when evaluated on a low number of samples. We compute it using features from the last convolutional layer of the Inception v3 model and considering only the RGB channels of the generated images. In order to evaluate the alpha mask generated by the RGBA generator, following [60], we employ masks

Table 1: Quantitative evaluation of our RGBA generator. We measure KID for instance generation quality, IoU (Jaccard) between the alpha masks and ICON segmentation, and CLIP Score for the caption/image similarity. †: our reimplementation, best results are highlighted in **bold**.

| | KID ↓ | IoU ↑ | CLIP Score ↑ |
|---|---|---|---|
| SD-v1.5 | 0.0839 | N.A. | 18.33 |
| PixArt-$\alpha$ | 0.0447 | N.A. | 18.10 |
| SD-v1.5 + MA | 0.0711 | 0.649 | 18.40 |
| PixArt-$\alpha$ + MA | 0.0558 | 0.811 | 18.03 |
| Text2Layer † | 0.0500 | 0.326 | 18.21 |
| LayerDiffusion | 0.0772 | 0.320 | 18.39 |
| Ours | **0.0150** | **0.892** | **18.49** |
| Ablation experiments | | | |
| No paired training | 0.0181 | 0.814 | 18.41 |
| No conditional sampling | 0.0152 | 0.887 | **18.49** |

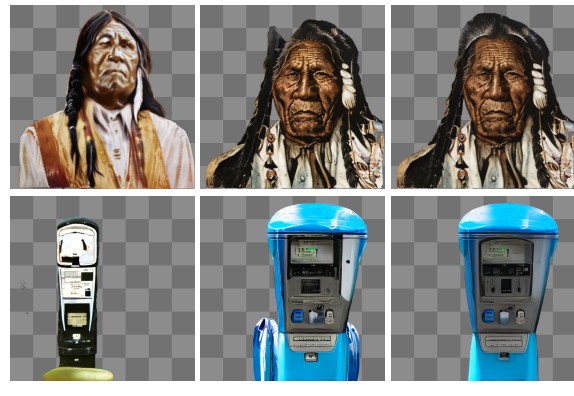

(a) Standard training (b) Standard sampling (c) Ours

Figure 4: Our proposed training and sampling approaches (c) improve results obtained with standard training (a) and standard sampling (b).

obtained with ICON with PVT backbone [61] as ground truth, and compute the Intersection-Over-Union (IOU) with the predicted alpha masks. Lastly, to evaluate the correlation between the captions and the content of the generated instances, we employ the CLIP Score [13], which computes the cosine similarity between image and text features using the CLIP model (ViT-B/16 variant) [39]. The results obtained are shown in Table 1. We can see that our approach obtains the best results for all the metrics, with KID of 0.0150, IoU of 0.892 and CLIP score of 18.49. PixArt-$\alpha$ has the second-best KID of 0.0447. However, the result is worsened when applying MA. Pixart + MA also has the second best IoU of 0.8111. SD-v1.5 has KID of 0.0839, result which is improved to 0.0711 by adding MA. The same model also achieves IoU of 0.649. Text2Layer has KID of 0.0500 and IoU of 0.326, while LayerDiffusion obtains KID of 0.0772 and IoU of 0.320. The latter's lower performance could be attributed to unreliable transparent generation, as instances can be generated without transparency due to the modelling strategy, affecting average performance. With regards to CLIP scores, they are relatively close to each other, with SD-v1.5 + MA obtaining the second-best score of 18.40.

**Ablation studies.** In order to explore the impact of our novel training procedure, we train two RGBA LDMs 1) without our conditioned training procedure (i.e. predicting noise $\epsilon$ without conditioning from different timesteps) and 2) without our conditional sampling inference where RGB and alpha are mutually conditioned for generation. Performance of these models is reported in Table 1, bottom rows. We can see that the largest gains in performance are obtained with our novel training procedure, while conditional generation results in more subtle improvement. As shown in Fig. 4, our conditional sampling allows to correct small details and critical areas of the alpha masks.

### 4.2 Scene Compositing results

**Generation details.** We build composite scenes in three steps: 1) RGBA instance generation, 2) layout building: we draw one bounding box per generated instance and rescale instances accordingly, 3) noise blending of generated instances according to a global prompt. We keep parameters consistent across all scene compositions unless specified otherwise: guidance scale $2.5$ (RGBA) and $4.5$ (Blending), guidance rescale $0.25$, noise blending steps $n = 30$, background blending $b = 20$ steps and consistency regularisation $n_s = 10$ steps. Both instances and composite image are generated over 50 steps. We use the Pixart-$\alpha$ model for compositing.

**Baselines.** We compare our approach to state of the art scene composition methods focusing on layout control and interactivity. Our first baseline is Pixart-$\alpha$ [6] to provide intuition into the complexity of requested prompts and limitation of relying solely on text controls. Next, we compare to GLIGEN [24], which learns additional cross attention layer to integrate bounding boxes and bounding box text descriptors as generation conditioning. Finally, we compare our approach to multi-layer methods MultiDiffusion [2] and contemporary approach Instance Diffusion [49] for completeness. Similarly to ours, these approaches rely on noise blending techniques to build composite scenes. Besides our

| Layout | Pixart-α | GLIGEN | MultiDiffusion | Inst. Diffusion | Ours |
| --- | --- | --- | --- | --- | --- |

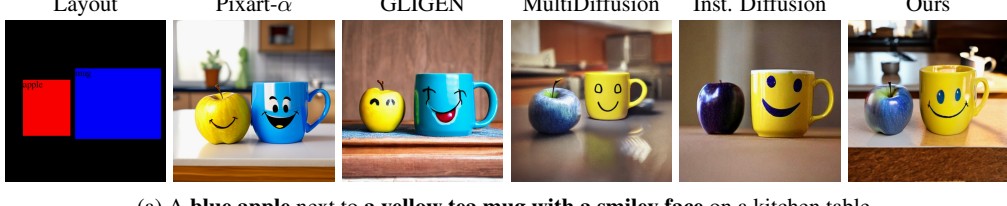

(a) A **blue apple** next to **a yellow tea mug with a smiley face** on a kitchen table.

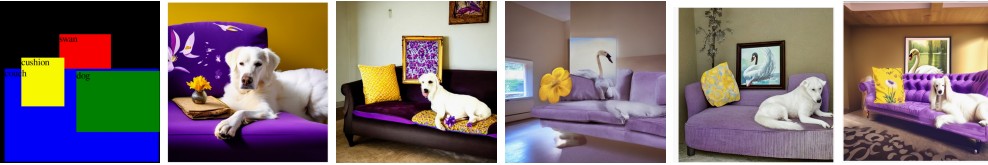

(b) **A white dog** lying on **a purple couch** next to **a yellow cushion with a flower pattern**, with **a swan painting** in the back.

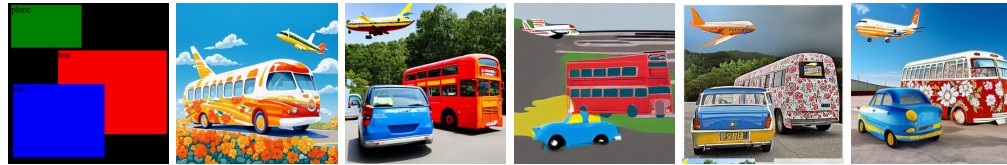

(c) **a blue and yellow car** in front of **a red bus covered with a white flower pattern** on the road with **a white airplane with orange stripes** in the sky.

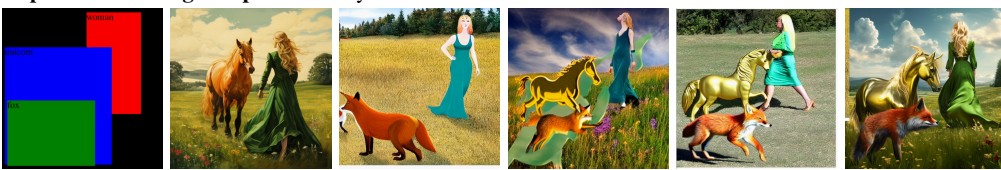

(d) **A red fox** in front of **a golden unicorn** and **a blonde woman in a long green dress walking away** in a meadow.

Figure 5: Visual examples of scene composition results. RGBA instances are highlighted in **bold**.

RGBA instance pre-generation, a crucial difference, in contrast to our multi-layer approach, is the use of noise averaging for overlapping objects. All methods use the same global prompt, GLIGEN and Instance diffusion leverage our bounding box layout and RGBA instance prompts as descriptors, while MultiDiffusion uses our instances alpha masks and captions as well.

Due to the inherently interactive nature of our and competing approach (layout design, instance ordering), we provide visual results in Fig. 5, comparing all methods across different scene compositions. For a fair comparison, we generate competing methods with multiple seeds and select the best result. We provide results for simple scenes with unusual attributes (a) and complex scenes with overlapping objects (b,c,d) (Additional results and ablations are available in Appendix E.2 and F.2). While GLIGEN is capable of accurately reproducing the desired layout, it often fails to assign the right attributes to objects (e.g. blue apple, orange plane, swan painting) and struggles with highly overlapping objects (fox and unicorn). In contrast, Multidiffusion is more accurate in terms of attribute assignments, but struggles to handle overlapping objects. This can be attributed to the noise averaging process, which fails to integrate a notion of instance ordering like our multi-layer approach. Instance diffusion achieves performance closest to ours, but still struggles with complex patterns, attributes and relative positioning (swan painting behind the couch, white bus with a red flower pattern, woman walking away). With our RGBA instance generation and multi-layer noise blending, we are able to accurately assign object attributes and follow the required layout, while successfully building smooth and realistic scenes.

**Scene manipulation results.** Finally, we evaluate our method's potential for scene manipulation. To optimise cross scene consistency, we set $b = 0$ (remove background blending) and set a common generation seed across all versions of the same scene. The manipulations we consider here are: attribute modification, instance replacement, and layout adjustment. We note that the first two tasks

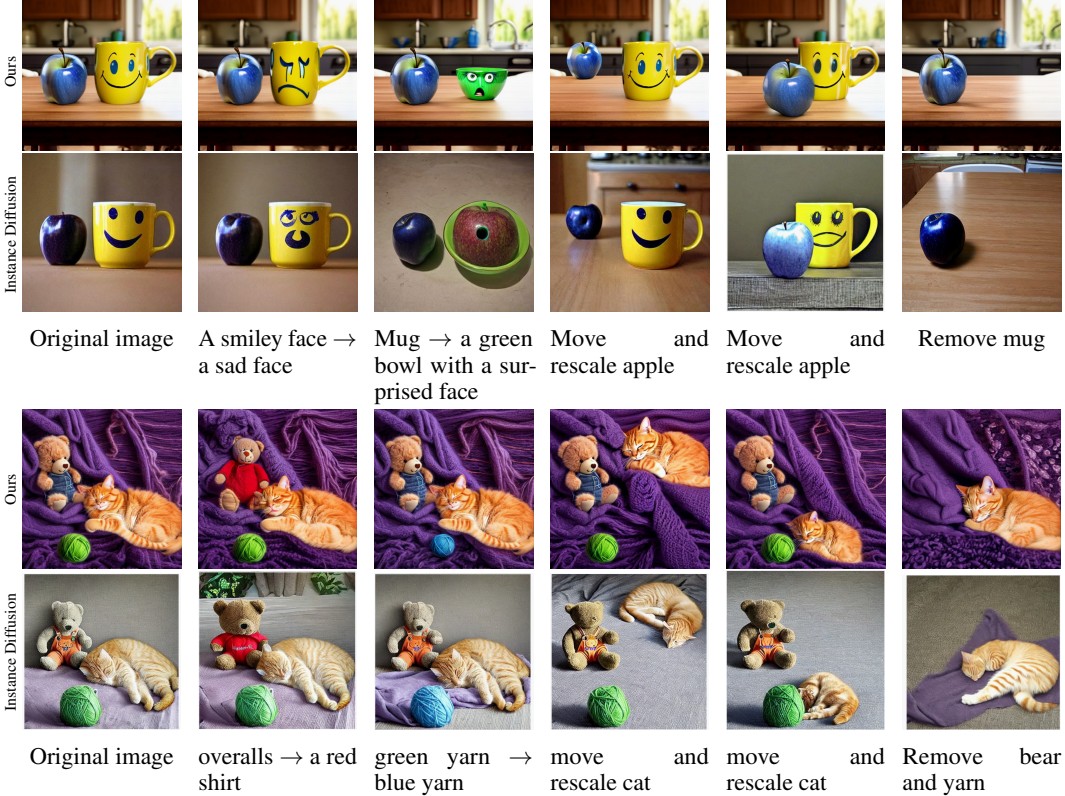

Figure 6: Visual examples of scene manipulations compared to Instance Diffusion. Our layer-based approach allows to replace instances or modify their positions.

require RGBA generation of new instances. We highlight that we *do not introduce any new explicit scene preservation or image editing technology* in this experiment, therefore evaluating our method's potential for scene manipulation and controllability. We compare our results to contemporary method Instance Diffusion, which has been highlighted to possess similar editing capabilities [49]. Results are reported in Fig. 6, showing that we are able to control and modify image content easily while maintaining strong consistency across different versions of the scene, without explicitly enforcing content preservation. This highlights the strong potential of multi-layer approaches to facilitate the development of image editing methods. We can see that we achieve substantially stronger scene preservation compared to instance diffusion, which can generate entirely different images and instances when modifications are too strong.

## 5  Conclusion

In this work, we introduced a novel multi-layer strategy to scene composition, that focuses on interactivity and fine-grained control. To achieve this, we proposed a new training paradigm that adapts diffusion models to generate transparent images, through channel disentanglement and conditional sampling. To build composite scenes from RGBA instances, we further present a multi-layer compositing strategy that concurrently generates increasingly complex scenes through cross-layer noise blending. Extensive experiments show that we are able to generate diverse, fine-grained RGBA instances, and assemble them in complex scenes, achieving precise control and high flexibility over scene structure. One key limitation of our approach is the independent generation of instances, increasing the challenge of assembling them in a coherent scene. Future work will explore conditioned RGBA generation to intrinsically generate coherent scenes, as well as RGBA editing methods to further improve fine-grained control over scene content.

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

# A RGBA training details and more experiments

## A.1 Datasets

The MuLAn [47] dataset consists of RGBA decompositions for over 44,000 RGB images from the LAION and COCO datasets. Developed through a training-free pipeline, MuLAn decomposes monocular RGB images into a series of RGBA layers, comprising of background and isolated instances. Our training RGBA images are instances that were automatically extracted from natural images, making this noisy but diverse dataset an excellent solution for a first fine-tuning step. In order to assemble the matting dataset, we employed data from AIM-500 [23], HIM-2k [46], Interactive Human Matting [10], RWP-636 [56], PPM-100 [19], AM-2k [22].

## A.2 Training details

Images were resized with bilinear interpolation then centre cropped to obtain a $512 \times 512$ image. We observed that this particular transformation normally preserved the majority of the content of the instances. The VAE was trained for a total of 23 epochs (with each epoch comprising of 42908 steps performed with a batch size of 2) employing the Adam optimiser with starting learning rate of $4.5e^{-6}$, $\beta_1 = 0.5$ and $\beta_2 = 0.9$. The discriminator adversarial loss was introduced after 50k steps with a weight of 0.5.

For the LDM fine-tuning we followed a similar pre-processing, but additionally performing horizontal flipping with $p = 0.5$. To train the model for classifier-free guidance, we dropped the caption conditioning with $p = 0.1$. We first perform a first stage of fine-tuning on MuLAn for 200k iterations and a second stage employing $50\%$ of the data from MuLAn and $50\%$ from the dataset obtained combining different matting datasets for 80k steps. This is because MuLAn is larger and offers a wider variety of subjects, while the matting dataset, primarily comprising humans and animals, offers finer, human-annotated alpha masks. In particular, we employed the AdamW optimiser [28] with a starting learning rate of $1e^{-5}$ and cosine scheduler. and trained in mixed precision with a batch size of 12. The VAE and LDM were initialised with the weights from PixArt-$\alpha$, while the text encoder (4.3B Flan-T5-XXL large language model) was kept frozen.

To maximise model expressiveness, detailed captions were automatically generated for instances using the Llava-NeXt model [26] with the following instruction: *"Write a detailed caption of image."*. Since the captioning model operates only in the RGB space, transparent pixels were included in the captions as descriptions of black backgrounds. We employed Phi-3 [1] to remove all such references by using the following instruction *"You are a talented ghostwriter given a text remove any mentions of the a black background while keeping the text as close as possible to the original."*.

Resizing and cropping training data can cause the generative model to also produce truncated instances. To limit this behaviour, inspired by [37], we store crop coordinates, indicating the pixels cropped from the top-left corner along the height $c_{top}$ and width $c_{left}$. These coordinates are then mapped with a sinusoidal embedding, similar to timesteps embeddings [6], which is followed by a two-layer MLP featuring SiLU activations. At inference time, we set $(c_{top}, c_{left}) = (0, 0)$. In addition, we employed Offset Noise, which improves the contrast in images created by the model [33]. At inference time, we observed improvements employing timestep trailing [25], classifier-free guidance [14] with the rescaling proposed by [25]. In particular, we observed the best results with guidance scale 2.5 and rescaling parameter 0.25 as shown in Sec. F.1.

For Text2Layer, we trained the CaT$^2$I-AE autoencoder and U-Net LDM following [60] on the MuLAn dataset, as the data used in [60] is not available. When sampling from PixArt-$\alpha$ and our model, we employed DPM-Solver [29], while for SD and Text2Layer we employed PNDM [27], default recommended parameters were used for LayerDiffusion.

## A.3 VAE Ablation

In Fig. 7, we visualise the impact of different training paradigms for our RGBA VAE on generated samples. We visualise images generated with a diffusion model fine-tuned on different VAE latent spaces. We can see that failing to disentangle the alpha and RGB channels (with a joint KL loss, or low regularisation weight) yields poor quality images with high contrast.

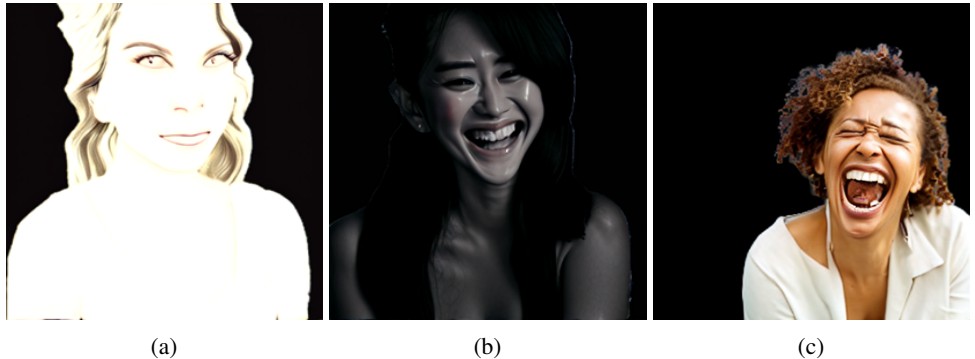

|       (a)       |       (b)       |       (c)       |

Figure 7: Images generated with our LDM fine-tuned in the latent space of VAEs that were trained with a single KL loss with weight $10e - 6$ (a), 2 separate KL losses each with weight $10e - 6$ (b), and 2 KL losses with weight 1 (c).

### A.4 Further Experiments

When fine-tuning the VAE, we also experimented with increasing the latent channels from four to eight in order to enhance the expressiveness of the latent space. However, we observed that while this modification improves the VAE reconstruction capabilities, it makes fine-tuning the LDM significantly more challenging and costly. Therefore, we decided to preserve the original latent dimension and encode both RGB and transparency information with our disentangled approach.

For the LDM, we experimented with parameter-efficient fine-tuning techniques such as LoRA [15], but observing an excessive decrease in image quality, we finally opted for full fine-tuning of the network.

## B Noise Blending Algorithm

To aid reader comprehension, we provide an algorithmic description of our multi-layer noise blending in Algorithm 1.

## C Broader and Societal Impact

Our work contributes positively to several domains. Artists, designers, and content creators can benefit from more precise control over their generated images, enhancing creativity and productivity. Additionally, the technology can be used in educational tools to help students and professionals learn about design, art, and computer graphics. Improved text-to-image generation can also make graphic design more accessible to individuals without extensive training in the field.

However, as with any generative model, there is a risk that the technology could be used to create disinformation, generate fake profiles, or produce harmful content. There is also a risk that generated images could be used to infringe on personal privacy, particularly if the model is misused to create realistic but fake images of individuals. Furthermore, the generated images could be used in ways that compromise security, such as creating counterfeit documents or misleading imagery for fraud.

To mitigate these risks, we propose several safeguarding measures. Releasing the model in a controlled manner can ensure that only verified and responsible users have access to it. Conducting thorough audits to ensure that the model does not perpetuate or amplify existing biases is also important. Maintaining transparency about the capabilities and limitations of the model, and establishing accountability protocols for its use, will help manage its impact. Providing resources and guidelines for ethical use, along with training for users on the potential societal impacts of the technology, can further promote responsible usage. Finally, developing features that can detect and prevent the creation of malicious content, such as watermarking techniques and usage restrictions, will enhance security measures.

---

**Algorithm 1:** Multi-layer scene composition

---

**Inputs**: $K$ pre-generated RGBA instances $\{I_k, M_k\}$,
A bounding box based scene layout $\mathcal{L} = \{[cx_k, cy_k, w_k, h_k], \text{for } k \in 1, \cdots, K\}$,
A latent diffusion model (VAE $\mathcal{E}$, diffusion model $\mathcal{D}$)
$T$ generation timesteps,
Random noise $\eta \in \mathcal{N}(0, I)$
Number of blending timesteps $n$,
background blending timesteps $b$,
cross-layer consistency timesteps $n_s$
**Output**: $K + 1$ multi-layer images (background, $K$ images with increasing number of instances)

*Generation set-up*
    Rescale $\{I_k, M_k\}$ according to $[cx_k, cy_k, w_k, h_k]$
    $x_k^0 \leftarrow \mathcal{E}(I_k)$
    $m_k \leftarrow$ Downsample $M_k$ to latent space dimension
    Compute $x_t^k$ at all diffusion timesteps $t \in [1, T]$ using DDIM inversion
    Initialise all images with same Gaussian noise: $y_T^k \leftarrow \eta$

*Composite scene generation*
    **For** $t = T$ **to** 0:          ▷ Loop over timesteps
        $\epsilon_\theta(y_t^k, t) \leftarrow \mathcal{D}(y_t^k, t)$     ▷ Predict noise update with diffusion model
        $y_{t-1}^k \leftarrow$ Update according to noise scheduler and $\epsilon_\theta(y_t^k, t)$
        **If** $t \geq b$:     ▷ Background blending optional step
            $y_{t-1}^0 \leftarrow y_{t-1}^0 \cdot m^* + \frac{1}{2}(y_{t-1}^0 + y_{t-1}^K) \cdot (1 - m^*)$
        **If** $t \geq n$:     ▷ Scene composition
            **For** $k = 0$ **to** K:
                $y_{t-1}^k \leftarrow y_{t-1}^{k-1} \cdot (1 - m_k) + x_{t-1}^k \cdot m_k$
        **Elif** $t \geq n + n_s$:     ▷ cross-layer consistency optional step
            **For** $k = 0$ **to** K:
                $y_{t-1}^K \leftarrow y_{t-1}^K \cdot (1 - m_k) + y_{t-1}^{k-1} \cdot m_k$

---

# D   Limitations

**RGBA generation.** Our RGBA generator is trained on a relatively small dataset, and doesn't leverage training data with intrinsic transparency information. We would expect substantial image quality improvement by increasing training data volume and quality. As our approach requires full fine-tuning of the diffusion model (including the VAE), interaction with pre-trained diffusion models is more challenging, as models operate in different latent spaces. Finally, our mutual conditioning sampling increases computation complexity at inference time. As our model was trained to handle different sampling strategies, we leave it up to the user to determine whether the fine-grained improvements observed are needed for generation tasks.

**Scene compositing.** By combining image inversion and multi-layer composition, we achieve much greater flexibility and control at the cost of increased computational and framework complexity. Future work will seek to simplify and automatise the generation pipeline, notably through automated instance prompt and layout generation. Generating instances independently can make scene composition challenging and require prompt engineering to obtain the right instance. Combining RGBA generation with techniques like ControlNet, or conditioning RGBA generation with background images has the potential to address these limitations.

# E  Additional Visual results

## E.1  RGBA Samples

In Fig. 8 we present additional examples of generated RGBA instances. Our model is able to successfully cover a broad range of subjects, including realistic portraits, stylised figures, inanimate objects, and fantasy characters.

## E.2  Scene Composition Results

In Fig. 9, we provide additional examples of scene compositions, highlighting our model's ability to successfully build complex scenes with fine-grained attributes and instance ordering. Limitations of competing methods are observed here as well, with multi-diffusion struggling with overlapping instances and relative positioning (couch image, flower vase image) and GLIGEN struggling to assign the right attributes to instances (turtle image, marble table in couch image). Contemporary method Instance Diffusion shows similar performance to ours here, as attributes and patterns are simpler.

For completeness, we additionally provide experiments with LayerDiffusion [57], aiming to generate similar scenes. However, the model does not allow to use layout to guide the generation process, with instances being generated based on background content only. Our experiments, using publicly available code, have shown that instances tend to be generated in the middle of the image, making complex composition difficult. Results are provided in Fig. 14.

# F  Parameter analysis

## F.1  RGBA generation: classifier-free guidance parameters

In order to find the best parameters for the classifier-free Guidance Scale (GS) and Guidance Rescaling (GR), we perform a grid search on validation data, measuring KID to evaluate image quality. As it can be observed in Fig. 10, we obtain the best result (lowest KID), with $GS = 2.5, GR = 0.25$

## F.2  Scene composition parameters

We provide a visual analysis of the impact of our key noise blending parameters on the generated image output. In Fig. 11, we visualise different images when modifying $n$ (number of timesteps where inverted instances noise is injected) and $b$ (number of steps where background noise is blended with the composite image). We can see that $n$ is the most crucial parameter: when $n$ is too low, no control over image content is achieved and the scene is solely controlled by the prompt input. Increasing $n$ increases faithfulness to the original RGBA instance appearance, at the cost of reduced blending quality. We observed that setting $n = 30$ yields the smoothest blending, maintaining instance attributes and positioning while at the same time adjusting appearance to fit the scene composition better.

Parameter $b$ has a subtler impact, adjusting the generated background image to blend with the scene layout more accurately. We can see notably in Fig. 11 that the blanket under the cat is adjusted and looks more realistic when $b > 0$. However, setting $b$ too high risks introducing instance information in the background area, strongly affecting the overall generated scene.

Next, we visualise the impact of parameter $n_s$, which enforces consistency between the composite scene and individual layers' instance areas in Fig. 12. For this experiment, we visualise intermediate layers as well, to highlight cross-layer consistency. We can see that setting $n_s > 0$ substantially improves consistency of instance appearance between individual instance layers and the final composite scene. While we considered introducing a similar consistency regularisation between individual layers, we observed reduced composition performance with this kind of strategy.

Finally, in Fig. 13 we visualise our method's consistency across multiple random seeds compared to competing methods Multidiffusion [2], GLIGEN [24] and Instance Diffusion [49]. By first generating instances and compositing them, we achieve strong scene consistency when modifying the random seed during our blending procedure. In contrast, GLIGEN and Instance Diffusion struggle to maintain attributes (including object types), while Multidiffusion often struggles with overlapping instances and their relative positioning.

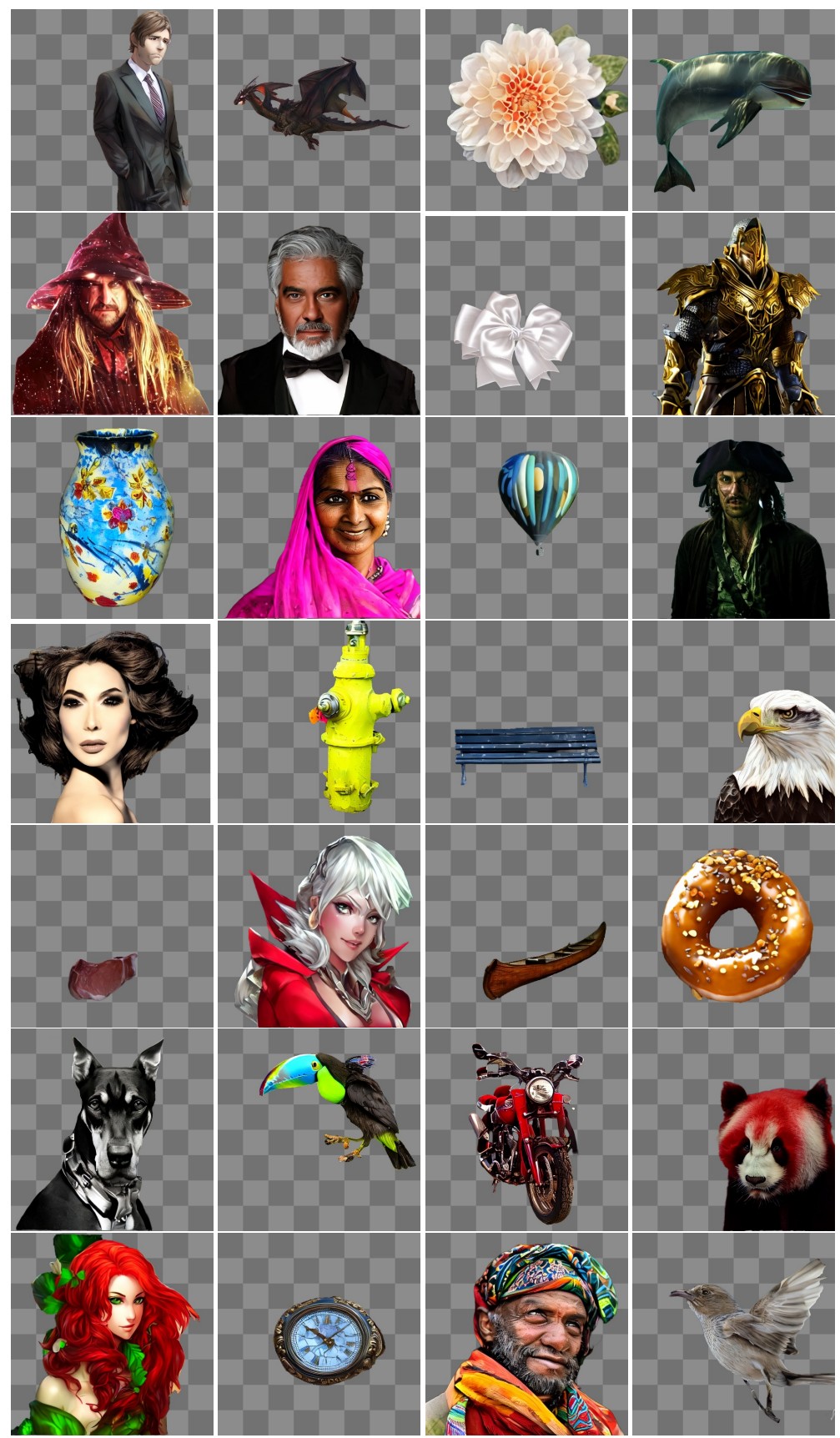

Figure 8: Sample RGBA instances. We are able to generate a wide variety of subjects.

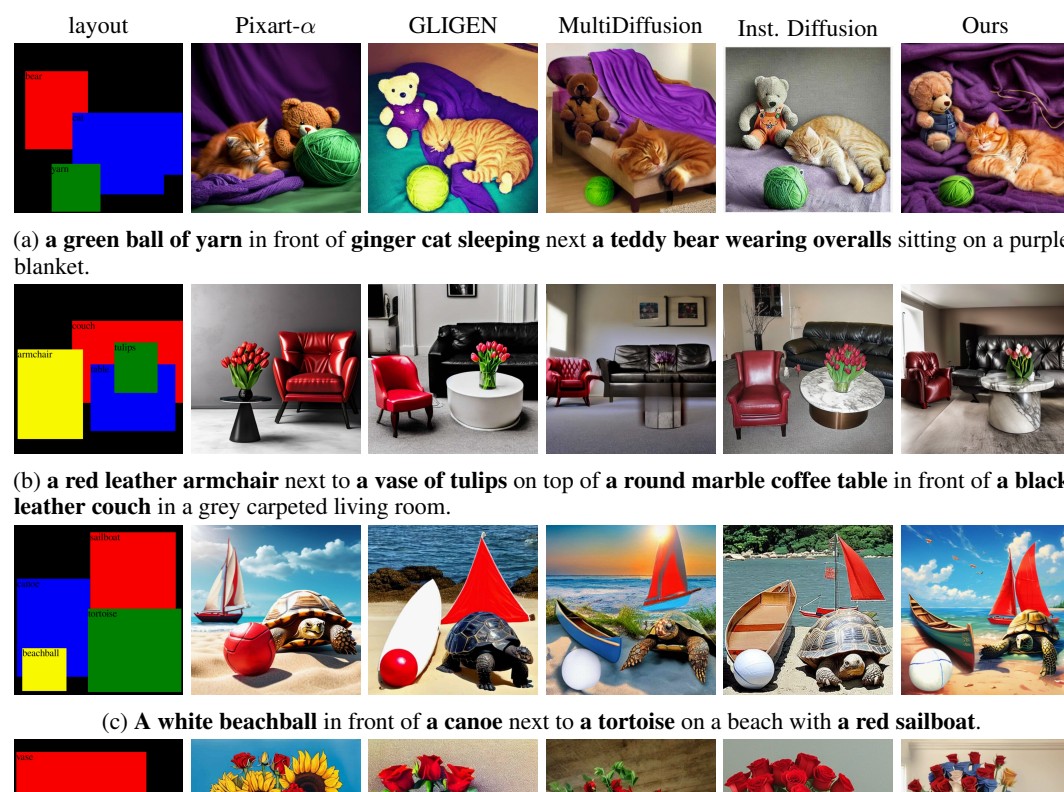

| layout | Pixart-$\alpha$ | GLIGEN | MultiDiffusion | Inst. Diffusion | Ours |
|---|---|---|---|---|---|

(a) **a green ball of yarn** in front of **ginger cat sleeping** next **a teddy bear wearing overalls** sitting on a purple blanket.

(b) **a red leather armchair** next to **a vase of tulips** on top of **a round marble coffee table** in front of **a black leather couch** in a grey carpeted living room.

(c) **A white beachball** in front of **a canoe** next to **a tortoise** on a beach with **a red sailboat**.

(d) **a book with a drawing of a sunflower on the cover** in front of **a bouquet of red roses in a blue vase**.

Figure 9: Additional scene composition results. RGBA instances and their attributes are bolded in the prompt.

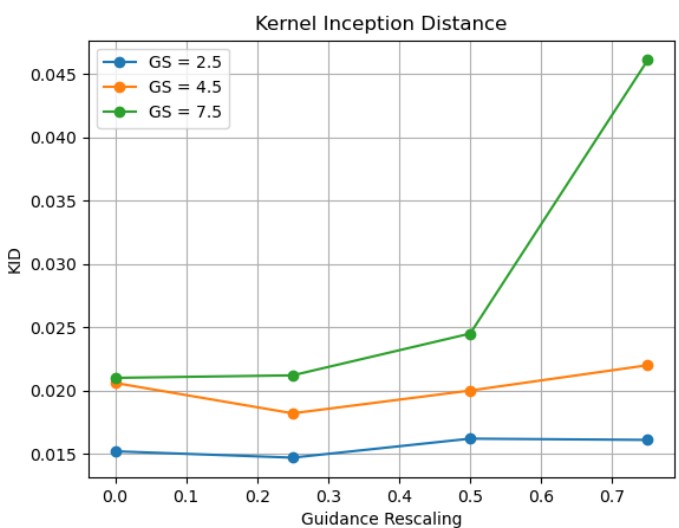

Figure 10: KID obtained with different combination of the guidance scale and guidance rescaling parameters. Setting $GS = 2.5, GR = 0.25$, we achieve the best results on validation data.

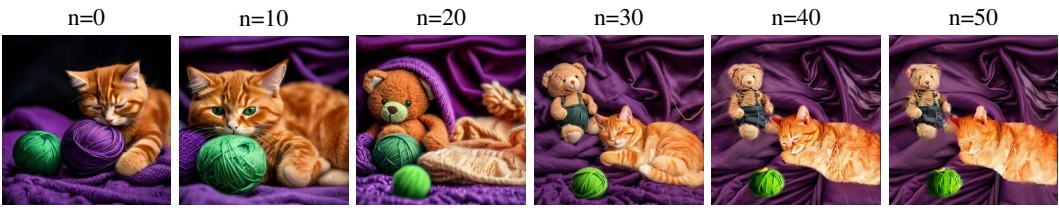

(a) Impact of adjusting instance blending parameter $n$ on scene composition. For this experiment, $b = 20$ and $n_s = 0$.



(b) Impact of adjusting background blending parameter $b$ on scene composition. For this experiment, $n = 30$ and $n_s = 10$.

Figure 11: Influence of scene composition parameters over generated scene content.

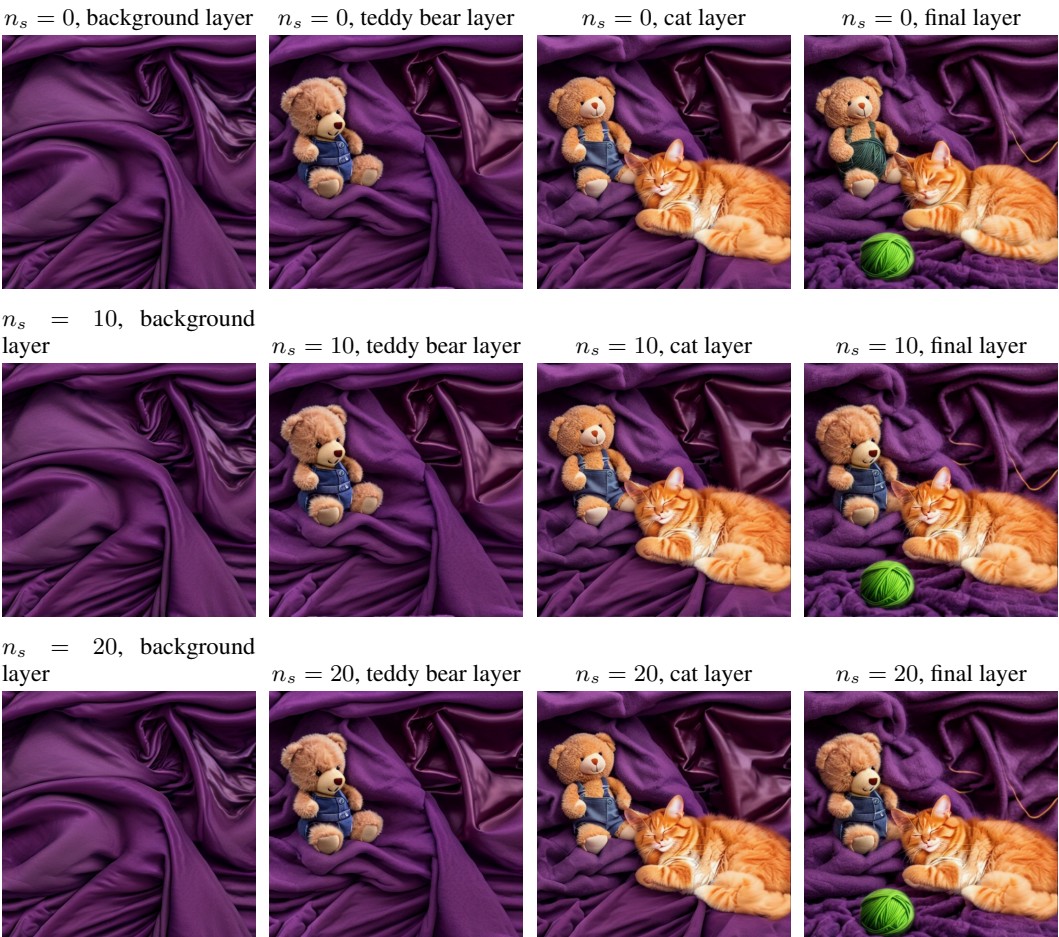

Figure 12: Influence of layer consistency parameter $n_s$ over all generated layers. For this experiment, we set $n = 30$ and $b = 20$.

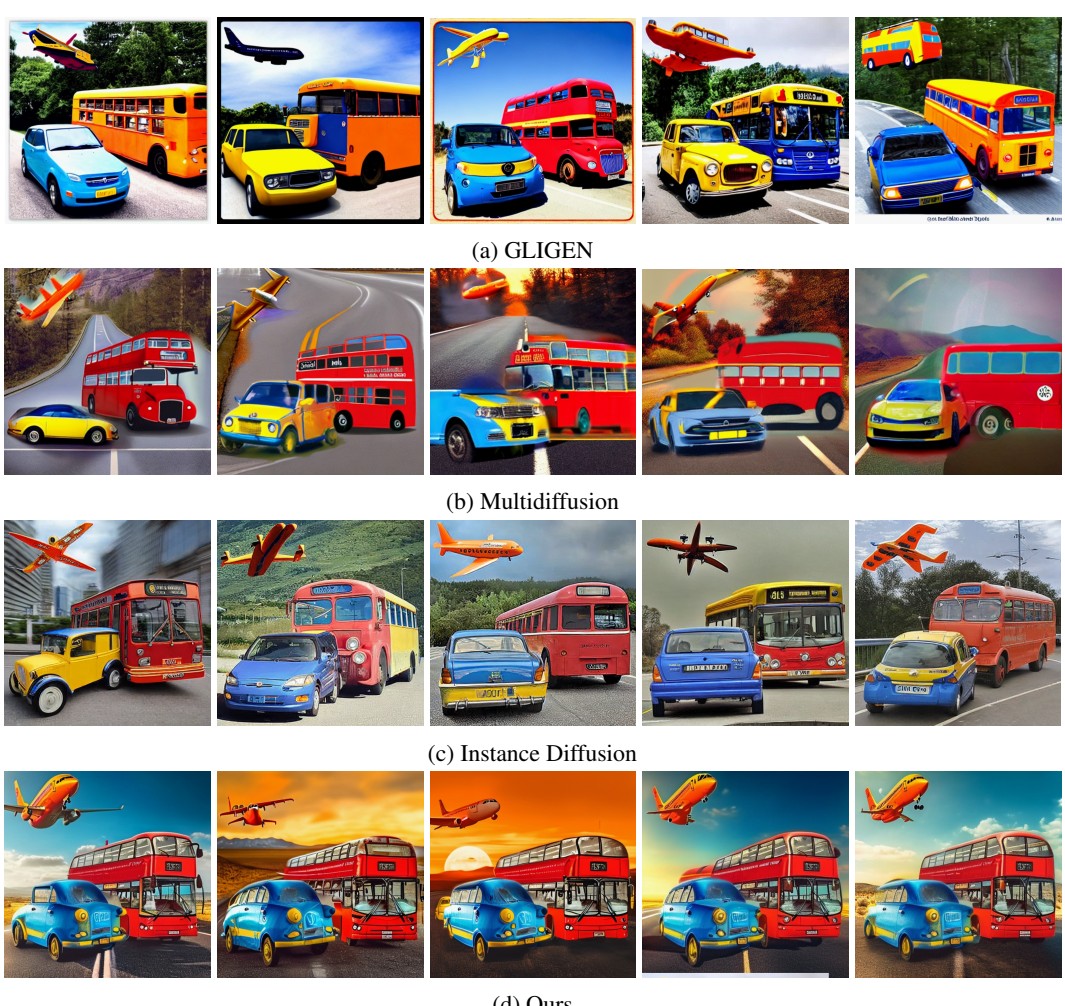

(a) GLIGEN

(b) Multidiffusion

(c) Instance Diffusion

(d) Ours

Figure 13: Impact of changing the random seed on scene composition consistency. Image caption: a blue and yellow car in front of a red bus on the road with an orange airplane in the sky.

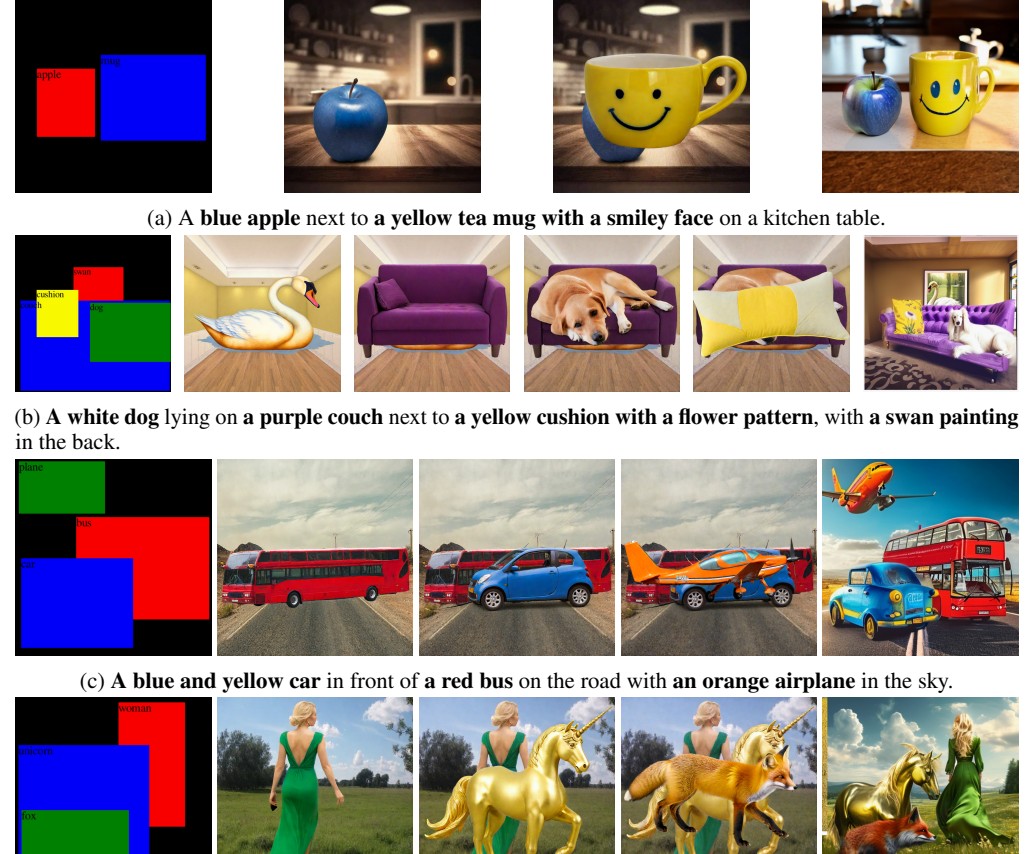

(a) A **blue apple** next to **a yellow tea mug with a smiley face** on a kitchen table.

(b) **A white dog** lying on **a purple couch** next to **a yellow cushion with a flower pattern**, with **a swan painting** in the back.

(c) **A blue and yellow car** in front of **a red bus** on the road with **an orange airplane** in the sky.

(d) **A red fox** in front of **a golden unicorn** and **a blonde woman in a long green dress walking away** in a meadow.

Figure 14: Visual examples of LayerDiffusion scene composition results. RGBA instances are highlighted in **bold**. From left to right: Layout image, LayerDiffusion images layer per layer, and our composite image.

