# OpenReview forum: "Generating compositional scenes via Text-to-image RGBA Instance Generation"
_NeurIPS.cc/2024/Conference — NeurIPS 2024 poster_

### Official Review · Reviewer_tSA3 · 2024-07-11

**Soundness:** 2
**Presentation:** 3
**Contribution:** 2
**Rating:** 4
**Confidence:** 4

**Summary:**

This study enhances text-based image generation diffusion models by introducing multi-layer noise blending and transparency-aware training procedures, enhancing control over the generated image content. This technique allows for finer control over the elements composing the image and improves the quality and consistency of the generated images through conditional sampling.

**Strengths:**

1. The approach allows for detailed control of object appearance and placement within images.
2. Images are produced with greater detail and consistency through the use of conditional sampling and transparency handling.
3. The method is straightforward and easy to understand.

**Weaknesses:**

1. Novelty: The field of layered-based image generation, which includes both depth-based and instance-based layering, has been thoroughly investigated. It remains unclear what theoretical or methodological innovations the author's approach offers compared to existing instance-layering methodologies. The capability for fine-grained control, as mentioned, seems achievable by most contemporary methods.

2. Evaluation Concerns: Recent developments in instance-based layering methods appear to have been neglected. An analytical discussion or empirical benchmarking against these newer methods is warranted. Pertinent literature includes:

 Huang, Runhui, et al. "LayerDiff: Exploring Text-guided Multi-layered Composable Image Synthesis via Layer-Collaborative Diffusion Model." arXiv preprint arXiv:2403.11929 (2024).

 Zhang, Lvmin, and Maneesh Agrawala. "Transparent image layer diffusion using latent transparency." arXiv preprint arXiv:2402.17113 (2024).

3. Experimental Assessment: There appears to be a lack of quantitative evaluations for GLIGEN and MultiDiffusion methodologies. It would be helpful if the authors could clarify why quantitative comparisons with these methods were not included. Additionally, presenting visual results for entire scenes from non-layer-based methods, such as the SD series models, would be valuable in assessing whether layout-independent methods enable more coherent or effective inter-instance interactions.

4. The independent generation of instances introduces significant complexity and poses challenges in assembling them into a coherent scene, as noted by the authors.

**Questions:**

How does the model manage unexpected object interactions or complex overlapping in scene compositing? Providing in-the-wild results that showcase complex object interactions and instance occlusions, such as scenarios depicting a person riding a motorcycle and a person wielding a wand, would help clarify the model’s capabilities in  intricate scene compositions.

**Limitations:**

The authors mention limitations briefly at the end of the conclusion.

---

> ### Author Rebuttal · Authors · 2024-08-06
>
> We thank the reviewer for their comments that increase the quality of our work, and address main
> concerns below.
>
> **Novelty:** while we agree with the reviewer that layered-based representation are receiving increasing
> attention, we argue that our proposed methodology has several unique components which paves
> the way for multi-layer innovation. Firstly, transparent image generation is an emerging topic, with
> very few related works dedicated to the task. To tackle this task, we propose a novel modelling
> strategy with a disentangled latent space and mutual conditioning. In addition, we provide several
> training insights to achieve better results with this type of generation task. Secondly, we disagree
> with the reviewer that the level of fine-grained control we are providing can be achieved with
> contemporary method, as evidenced by our experiments. While common layered methods allow
> control over location and appearance of objects, they often struggle with patterns and fine-grained
> instance details, as all components are generated at the same time. We have investigated competing
> methods with publicly available implementations, and consistently observed difficulties in achieving
> the right level of details, and/or accurate instance positioning for the type of complex prompts we
> are showing in our manuscript. We highlight that we are working with substantially more complex
> image descriptions than e.g. T2I Compbench (reference [16]). In contrast, by generating instances
> separately, we can control fine-grained attributes and focus on scene coherence at blending time.
> Lastly, our multi-layer scene composition is also different from pre-existing methodology, due to our
> pre-generated instances. Here, instead of focusing on generating the right instance prior to a global
> merger, we focus on instance integration within the scene, through iterative instance addition layer
> by layer, at each timestep. This allows more natural integration of individual instances in the scene.
>
> **Related works:** While we focused our evaluation on published work, we agree with the reviewer
> that LayerDiffusion [1] is highly relevant recent related work, and update our manuscript to add
> comparisons to this work, both in related works and experiments. We highlight that this work
> was developed in parallel to ours, and our methodology and handling of transparency is entirely
> different. Furthermore, we highlight that the multi-layer scene generation process of LayerDiffusion
> is substantially more restricted, as instance location cannot be specified and is solely controlled
> by background appearance. We provide quantitative and qualitative comparisons in this rebuttal.
> Experimental details, and discussion are available in the global rebuttal.
> LayerDiff [2] is contemporaneous work to ours, and is closer to a multi-layer extension to Text2Layer
> than our work. We note an important limitation of this work, which is that occluded components are
> not generated. This makes scene manipulation tasks more challenging, and substantially reduces flexibility. Unfortunately, due to the complex dataset building required and lack of publicly available code,
> we are not able to provide comparisons to this work, but will integrate it in our related works section.
>
> **Experiments:** We agree with the reviewer that quantitative experiments would be a great addition.
> Unfortunately, as discussed in sec 4.2 (Baselines), quantitative evaluation is highly challenging due
> to the interactive nature of our methodology, as well as the highly complex nature of image prompts
> we are working with. This would require the development of a novel benchmark, which we leave
> to future work as quantitative evaluation remains an open problem for controllable generation tasks.
> We have made efforts to provide many examples highlighting different properties and benefits of
> our work, and to thoroughly analyse our method’s behaviour in our supplementary experiments. We
> note that we do provide a non-layered baseline with the PixArt-alpha model (reference [6]) and are
> happy to additionally provide a Stable Diffusion baseline as well if needed.
>
> **Instance interactions:** We agree with the reviewer that independent instance generation is a
> limitation of our work, and an important focus of our future work. Our method in its current state can
> mitigate these issues with precise instance prompts, and stronger blending freedom at composition
> time. Highly intricate interactions can be achieved without altering our main methodology, by
> fine-tuning our RGBA generator with higher quality, fine-grained training data, or making it capable
> of generating grouped instances.
>
> [1] Zhang, Lvmin, and Maneesh Agrawala. "Transparent image layer diffusion using latent
> transparency." arXiv preprint arXiv:2402.17113 (2024).
>
> [2] Huang, Runhui, et al. "LayerDiff: Exploring Text-guided Multi-layered Composable Image
> Synthesis via Layer-Collaborative Diffusion Model." arXiv preprint arXiv:2403.11929 (2024).

---

### Official Review · Reviewer_S8j5 · 2024-07-12

**Soundness:** 3
**Presentation:** 3
**Contribution:** 2
**Rating:** 4
**Confidence:** 4

**Summary:**

This paper introduces a novel method for generating complex images from textual descriptions with a high degree of control over object attributes and scene composition. They introduce a new training paradigm for adapting a diffusion model to generate isolated objects with transparency, using a disentangled representation for RGB and alpha channels. Moreover, a multi-layer scene compositing strategy based on noise blending is developed, enabling the generation and manipulation of complex scenes from multiple instance images. Overall, the task is interesting and the proposed method achieves fantastic generation ability.

**Strengths:**

1. The paper presents a framework for fine-grained controllable generation, providing user control over instance appearance and location with intrinsic scene manipulation capabilities. The results show superior controllability of their approach.

**Weaknesses:**

1. I think this paper ignores a similar and pioneer work: LayerDiffusion [1], which achieves the RGBA generation by finetuning the VAE and diffusion models. It is necessary that this paper should compare with this prior work.
2. It seems that the Multi-Layer Noise Blending would cause unnatural image generation. I hope you can provide more discussion about the composition of foreground and background.
3. A deep and clear literature review of prior works is very important for an excellent work. More discussions about the LayerDiffusion[1] should be included in this paper.

[1] Zhang L, Agrawala M. Transparent image layer diffusion using latent transparency[J]. arXiv preprint arXiv:2402.17113, 2024.

**Questions:**

None

---

> ### Author Rebuttal · Authors · 2024-08-06
>
> We thank the reviewer for their comments that improve the quality of our work. We are grateful
> for the positive remarks (interesting work, fantastic generation ability, superior controllability), and
> address the reviewer’s main concerns: comparison to LayerDiffusion and the impact of multi-layer
> noise blending on realistic generation.
>
> **LayerDiffusion:** Firstly, we highlight that our work was developed in parallel to LayerDiffusion [1],
> which is nearly contemporaneous to ours, and our methodological decisions were not inspired from
> this work. This can be evidenced by our entirely different methodological decision : we explicitly
> model the transparent channel, and disentangle the latent space; while LayerDiffusion implicitly
> incorporates transparent information. However, due to the very similar objective of transparent
> generation, we agree with the reviewer that comparisons are important, and extend our evaluation
> to include a comparison to LayerDiffusion. Experimental details, results and discussion are available
> in the global rebuttal.
>
> Besides transparent generation, LayerDiffusion also offers multi-layer generation properties by
> generating instances conditioned on a background image. We point out here that our blending strategy
> affords substantially better controllability, as LayerDiffusion does not allow to specify the location
> of these instances (see Figure 2 of this rebuttal). We will update our manuscript to incorporate
> discussions on LayerDiffusion in related work, and add experiments provided in this rebuttal.
>
> **Multi-layer noise blending** can lead to unnatural generation, as evidenced in our supplementary
> experiments (see Supp. F.2 and Figure 12), when noise injection constraints are too strong. Our
> method is designed with this challenge in mind, with two important components to ensure natural
> looking images: 1) iterative integration of instances in the global scene, and 2) imposing constraints
> in early timesteps only. The latter ensures naturalness of generated scenes by leveraging the intrinsic
> biases of diffusion models. Imposing constraints only at early timesteps ensures instance locations
> and key attributes are integrated, while giving enough freedom to the model to generate realistic
> images. The impact of increasing or decreasing constraints is discussed in more details in our
> supplementary section F.2. The former component, unique to our multi-layer approach, iteratively
> integrates new instances in separate layers. Generating separate images allows to smoothly integrate
> each instance in the scene individually, simplifying the task in comparison to methods that aim to
> blend all instances in the scene at once.
>
> [1] Zhang, Lvmin, and Maneesh Agrawala. "Transparent image layer diffusion using latent
> transparency." arXiv preprint arXiv:2402.17113 (2024).

---

> > ### Comment · Reviewer_S8j5 · 2024-08-13
> >
> > Thanks for the response!
> >
> > I still think that the novelty is limited and the experimental results are not very convincing. I will keep my score.

---

> > > ### Author Response · Authors · 2024-08-13
> > >
> > > Dear reviewer, we thank you for your quick response. We have made efforts to address all your comments: we have provided a comparison to LayerDiffusion where we clearly outperform the method quantitatively with markedly different methodology, and have thoroughly discussed the limitations of multi-layer noise blending and our key innovations to address them. We are not aware of alternative works proposing to pre-generate RGBA instances for multi-layer scene composition with fine-grained control. Please let us know what aspects of our work remain unsatisfactory so we can address them.

---

### Official Review · Reviewer_wSp1 · 2024-07-13

**Soundness:** 3
**Presentation:** 3
**Contribution:** 3
**Rating:** 7
**Confidence:** 4

**Summary:**

This paper proposes to use a diffusion model to generate separate objects and then apply multi-layer noise blending to build a composite scene. A RGBA generator is finetuned from a latent diffusion model to generate alpha transparency for objects in addition to RGB. A transparency-aware training procedure is designed for both VAE and diffusion models. Then the generated objects are iteratively injected into a scene with a background according to a specific layout.

**Strengths:**

1.	To adapt the VAE and diffusion model for RGBA transpancy generation, they make several key changes in training and model, such as using disentangling latents. These changes are nontrivial.
2.	Proposed a novel background blending in addition to multi-layer composition process, which ensures the consistency between objects and the background.
3.	Extensive experiments have been conducted on state-of-the-art models and have shown good results.

**Weaknesses:**

The multi-layer scene composition method requires generating K + 1 images for composition, which is inefficient. Since RGBA objects have already been generated, there might be more efficient ways to fuse them in one step or reduce the cost of each iteration.

**Questions:**

A more detailed algorithm for 3.3 might be necessary to clarify how all the mechanisms work together, since the diffusion processes in the first n steps and b steps are different.

---

> ### Author Rebuttal · Authors · 2024-08-06
>
> We thank the reviewer for their comments that increase the quality of our work and are encouraged
> by the positive feedback. As discussed in our limitation section in supp. D, we agree that our
> more expensive generation is a limitation of our work. However, we expect this cost to be heavily
> amortised once a critical mass of RGBA assets has been generated. The same principle is also the
> current practice for 3D digital assets. We further highlight that it is a common limitation of layered
> representations, where layers require separate generation processes, and consider that the added
> flexibility and blending quality can outweigh additional generation cost. Nonetheless, we intend
> to explore more efficient solutions in future works, notably by exploring ways to combine instance
> generation and blending while maintaining fine-grained control.
>
> We thank the reviewer for highlighting clarity issues in section 3.3. We will update our manuscript
> to include the following algorithmic description of the section in supplementary materials to facilitate
> understanding.
>
> Multi-layer scene composition algorithm:
> > **Inputs**: $K$ pre-generated RGBA instances $\{I_k,M_k\}$,
> A bounding box based scene layout $\mathcal{L}=\{[cx_k, cy_k, w_k, h_k ], \text{for } k \in 1,\cdots,K\}$,
> A latent diffusion model (VAE $\mathcal{E}$, diffusion model $\mathcal{D}$)
> T generation timesteps
> Random noise $\eta \in \mathcal{N}(0,I)$
> Number of blending timesteps $n$,
> background blending timesteps $b$,
> cross-layer consistency timesteps $n_s$
> **Output**: $K+1$ multi-layer images (background, $K$ images with increasing number of instances)
>
> > *Generation set-up*
>         >>Rescale $\{I_k,M_k\}$ according to $[cx_k, cy_k, w_k, h_k ]$
>         $x^0_k \leftarrow \mathcal{E}(I_k)$
>         $m_k \leftarrow$ Downsample $M_k$ to latent space dimension
>         Compute $x^k_t$ at all diffusion timesteps $t \in [1,T]$ using DDIM inversion
>          Initialise all images with same Gaussian noise: $y^k_T \leftarrow \eta$
>
> >*Composite scene generation*
>        >>**For** $t=T$ **to** 0:  *Loop over timesteps*
>         >>> *Predict noise update with diffusion mode*l: $\epsilon_{\theta}(y^k_t,t) \leftarrow \mathcal{D}(y^k_t,t)$
>        $y^k_{t-1} \leftarrow$ Update according to noise scheduler and $\epsilon_{\theta}(y^k_t,t)$
>        **If** $t \geq b$: *Background blending optional step*
>        >>>> $y^0_{t-1} \leftarrow$ $y^0_{t-1} \cdot m^*+\frac{1}{2} (y^0_{t-1}+ y^K_{t-1})\cdot (1-m^*)$
>
>  >>>**If** $t \geq n$: *Scene composition*
>  >>>>**For** $k=1$ **to** K:
> >>>>> $y^k_{t-1} \leftarrow$  $y_{t-1}^{k-1} \cdot(1-m_k)+x^k_{t-1}\cdot m_k$
>
> >>>**Elif** $t \geq n+n_s$: *cross-layer consistency optional step*
> >>>>**For** $k=0$ **to** K:
> >>>>> $ y^K_{t-1} \leftarrow y^K_{t-1} \cdot(1-m_k)+y_{t-1}^{k-1} \cdot m_k$

---

> > ### Comment · Reviewer_wSp1 · 2024-08-14
> >
> > Thanks for the response. I think this is an interesting work. I will keep the current rating.

---

### Official Review · Reviewer_GbJP · 2024-07-17

**Soundness:** 3
**Presentation:** 3
**Contribution:** 1
**Rating:** 4
**Confidence:** 3

**Summary:**

This paper describes a diffusion model based method for layered text-to-image generation with RGBA masks (transparency information). This is a useful approach when aiming to generate complex images with many objects. To that end, the base VAE as well as the latent diffusion model are adapted to handle another channel and fine-tune to a corresponding dataset. The method is evaluated against base models and visual results show convincing results.

**Strengths:**

- The method is sound and well-engineered for this problem using sota building blocks.
- The paper is well-written and structured.
- It is easy to follow the paper.
- The visual results are convincing that the method works.

**Weaknesses:**

My main concern is regarding evaluation. While the method demonstrates that it works (visually), and quantitative evaluation shows that it performs better than off-the-shelf text-to-image models, there is a complete lack of comparison against methods that are more relevant to that particular task. The paper mentions many related works, but evaluation is only against base models, and comparisons to alternative methods are missing.
  - Naive Text2Layer extension to multiple layers instead of just two
  - No editing / inpainting methods (discussed in related work) are used for comparison
  - LayerDiff https://arxiv.org/abs/2403.11929
  - No comparison to controllability methods such as ReCo, BoxDiff etc.
  - No comparison to Layered rendering diffusion model for zero-shot guided image synthesis
Because of this, it is difficult to judge the technical contributions and novelty of the proposed method. Mainly because it seems like a straightforward adaptation of an existing pipeline through fine-tuning. More justification and experimental evidence are required to demonstrate and ablate the necessary design choices to go from an existing backbone to the proposed method.

**Questions:**

-

**Limitations:**

-

---

> ### Author Rebuttal · Authors · 2024-08-06
>
> We thank the reviewer for their comments that increase the quality of our work. We are encouraged
> by the positive comments that our work is useful, our methodology is sound, and our visual results
> are convincing. We now address the main limitations raised: 1) limited contribution and novelty,
> and 2) experimental comparisons.
>
> **Novelty:** we have aimed to design a novel controllable generation method with fine-grained control
> and interactivity in mind. To this end, we propose to first generate RGBA assets, and then assemble
> them in a composite scene. Generating images with transparency information is far from trivial (as
> mentioned by RwSp1) as the concept of transparency is foreign to diffusion models. We introduce
> novel modelling and training methodology for this purpose. In contrast with LayerDiffusion [1],
> which injects transparency information in the null space of the VAE to facilitate fine-tuning, we
> explicitly model transparency through disentanglement of RGB and alpha channels. Explicitly modelling transparency allows for more precise and nuanced generation of RGBA images, and guarantees
> that we generate images with a transparent background, which is not the case with LayerDiffusion.
> Text2Layer (reference [55]) is a substantially different approach as well, requiring triplets of background, foreground and saliency maps for training; and predicting all components together. Crucially,
> this approach doesn’t generate individual instances, but a foreground comprising all instances.
> Our scene composition leverages these RGBA assets in a novel multi-layer approach. In contrast
> with existing multi-layer methods which generate layers in parallel before merging them, we build
> a multi-layer composition process where each layer is dependent on the lower image layer, by
> integrating new instances in an increasingly complex image one layer at a time. This process allows
> to smoothly integrate each component individually (rather than trying to blend all instances together
> at once) and is uniquely afforded by our RGBA asset pre-generation. This further allows scene
> manipulations very easily, as pre-generated RGBA instances can easily be moved, replaced, and
> have their appearance frozen.
>
> **Evaluation:** We have made efforts to compare to highly relevant, widely used, state of the art works,
> for both RGBA generation and scene composition, at the time of submission. For the former, we
> reimplemented Text2Layer as the closest published work capable of RGBA generation. We highlight
> that extension of this work to multiple layers is far from straightforward, as disentanglement of
> foreground instances through their saliency segmentation approach is non trivial, and is beyond the
> scope of this work. We agree with the reviewer that nearly contemporaneous work LayerDiffusion
> is an important comparison, and provide quantitative and qualitative comparisons in this rebuttal.
> Experimental details, results and discussion are available in the global rebuttal.
>
>
> Regarding scene composition evaluation, we have made efforts to compare to open-source, widely
> used state of the art works designed for compositional generation. As discussed in section 4.2 (Baselines), we have compared to layout-based generation method GLIGEN, which in our experiments, more consistently outperformed boxdiff and ReCO (references [47] and [50]) and therefore
> constituted our best baseline bounding box based method. Similarly, for multi-layer scene generation,
> we have chosen multidiffusion, due to its strong performance and available implementation.
> Unfortunately, we were not able to reimplement Layered rendering diffusion model for zero-shot
> guided image synthesis (LRDiff, reference [36] in our manuscript) for comparison, due to missing
> implementation details and no public code. We highlight that LayerDiff[2] is contemporaneous work
> without available opensource code. Due to the expensive data construction and training costs, we are
> unable to provide comparisons to this work. While our focus is on generating all image components,
> including occluded areas, LayerDiff only generates visible pixels. This in turns, strongly limit scene
> manipulation abilities, which is one of the key benefits of our representation.
> Finally, while inpainting and editing techniques are relevant related work, we highlight that the focus
> of our work is not editing, but interactive and controllable scene generation. Our scene manipulation
> experiments highlight intrinsic benefits of using RGBA assets to easily manipulate scene content
> with strong content preservation. However, we do not aim to directly compete with editing methods,
> as we have not introduced any explicit editing mechanisms.
>
> [1] Zhang, Lvmin, and Maneesh Agrawala. "Transparent image layer diffusion using latent
> transparency." arXiv preprint arXiv:2402.17113 (2024).
>
> [2] Huang, Runhui, et al. "LayerDiff: Exploring Text-guided Multi-layered Composable Image
> Synthesis via Layer-Collaborative Diffusion Model." arXiv preprint arXiv:2403.11929 (2024).

---

> > ### Comment · Reviewer_GbJP · 2024-08-13
> >
> > Thanks for the respones. I have updated my score to BR because I still think that the technical contributions are limited and the experimental comparisons are not convincing enough to lean towards acceptance.

---

> > > ### Author Response · Authors · 2024-08-13
> > >
> > > Dear reviewer, we thank you for upgrading your score. We have provided comparisons with all relevant state of the art open-source works, including now LayerDiffusion which we outperform, and discussed all modern closed source works in our related works. Our adaptation to transparent generation is entirely novel with no alternative works proposing disentangled latent spaces and mutual conditioning. Our second contribution, our multi-layer blending method, proposes novel ideas such as our iterative layer-wise instance integration process.  Please let us know what aspects of our work remain unsatisfactory so we can address them.

---

### Author Rebuttal · Authors · 2024-08-06

We thank the reviewers for their detailed reviews, insights and comments that improve the quality of
our work. We are encouraged by the positive comments: the soundness of our method (RGbJP), the
quality of our results (RGbJP, RwSp1, RS8j5), the non triviality of our work (RwSp1), and detailed
control afforded by our method (RtSA3). We have made efforts to address all reviewers concerns in
individual rebuttals, through additional experiments and clarifications. The reviewers main concerns
were comparison of our method with LayerDiffusion [1] and novelty of our work. We restate our
key innovations below, and discuss our experimental details and results on LayerDiffusion.

**Novelty:** We designed our approach with interactivity and controllability in mind. Our key objective
is the development of a controllable generation solution were the user can easily build complex
scenes, control instances appearances, and adjust scene layout and content. In particular, we
are considering scenes with complex layouts, where multiple instances have unique patterns and
attributes, going well beyond compositional benchmarks such as T2I-Compbench (reference [16]).
Our key idea is the use of pre-generated RGBA assets, and their composition through a multi-layer
process. This allows precise attribute control, ensures that the blending process focuses on scene
coherence, and substantially facilitates scene manipulation tasks. To achieve this, we introduce 1) a novel modelling and training approach for generation of transparent images, and 2) a novel multi-layer scene composition method, that blends pre-generated assets into a scene. Integrating instances one by one in the scene allows for smoother scenes where relative position of instances in 3D space (A in front of B) is clearly established and easily controlled. Our experiments have shown that our approach allows a degree of fine-grained control that competing methods cannot achieve.

**LayerDiffusion:** While we focused our original manuscript on comparison with published, state
of the art work, we agree that the recent LayerDiffusion is highly relevant work, and now provide
comparisons with this related work in our rebuttal which will be added to our manuscript. We
ran experiments using the official implementation and recommended parameters [3], we used the
Juggernaut XL V6 SD XL model with 20 Steps, DPM++ 2M SDE sampler and a CFG scale of
5. For RGBA evaluation, instances were generated using the Foreground Model model with the
same captions as for the other models. For the scene composition visualisation we have used the
Background to blended image model where we iteratively add instances to the previously blended
image, following the protocol in [1] ( Figure 9). Images were generated across multiple seeds, and the
best results was selected. Results are available in the attached PDF document. Figure 1 shows visual
comparisons of generated instances, showing that while LayerDiffusion is able to generate high
quality instances, it does not necessarily respect prompt details (for example, the impressionist style).
Quantitative results in Table 1 show that we outperform LayerDiffusion on all metrics. The lower
IoU performance could notably be attributed to the implicit transparency modelling, which does
not guarantee that accurate transparency layers are generated. Lastly scene composition experiments
in Figure 2 show that the model struggles with iterative integration of instances, with a strong bias
towards the centre of the image, and no ability to control where instances are added.
We restate key differences between our work and LayerDiffusion: while both train a model capable
of generating RGBA images, our RGBA modelling approach is entirely different, where we build
a disentangled latent space and explicitly model interactions between RGB and Alpha channels.
In contrast, LayerDiffusion encodes transparency within the null space of the VAE, so as to train
a diffusion model following standard practice. We further highlight that our scene composition is
substantially more flexible, as LayerDiffusion only generates instances conditioned on a background
image, with no control over instance positioning. As shown in Figure 2 of our attached results, this
can lead to poorly positioned instances, with a bias towards the centre of the image.

**Baselines:** The field of controllable generation is very popular, and there exist many competing works
aiming for similar objectives. Considering the interactive nature of our work, providing a reliable
quantitative evaluation of our results is highly challenging and an open problem. For qualitative
evaluations, we have made efforts to select the strongest open-source baselines, covering standard
image generation (PixArt-alpha (reference [6])), layout based generation (GLIGEN (reference [23])),
and multi-layer (multi-diffusion (reference [2])), and attempted to reproduce alternative closed source
methods. We now additionally add LayerDiffusion as a highly relevant baseline. We will additionally
enrich our related work to discuss contemporaneous, closed source, work LayerDiff[2]. This work
focuses on predicting different image regions separately with dedicated prompts, and in contrast
to ours, doesn’t predict occluded areas. The methodology, closer to Text2Layer (reference [55]),
requires expensive dataset building through instance segmentation.

[1] Zhang, Lvmin, and Maneesh Agrawala. "Transparent image layer diffusion using latent
transparency." arXiv preprint arXiv:2402.17113 (2024).

[2] Huang, Runhui, et al. "LayerDiff: Exploring Text-guided Multi-layered Composable Image
Synthesis via Layer-Collaborative Diffusion Model." arXiv preprint arXiv:2403.11929 (2024).

[3] https://github.com/lllyasviel/sd-forge-layerdiffuse

---

### Decision · Program_Chairs · 2024-09-25

**Decision:**

Accept (poster)

**Comment:**

This paper proposes a text-to-image (T2I) diffusion model (DM) to generate separate objects (with RGBA masks to represent transparency information). The proposed method can generate complex images with a high degree of controllability (e.g, instance appearance and location).

The reviewers have mixed opinions. The significant concern of the reviewers is the lack of novelty of the proposed method. More specifically, three reviewers pointed out the lack of comparison with LayerDiff [1] and LayerDiffusion [2]. Although this paper and the previous works [1,2] have a high similarity, I can't entirely agree with the reviewers' opinion that these works should be compared with the proposed method at the submission time. Both of them were uploaded to ArXiv this year (Feb and Mar), and the submission deadline for NeurIPS was May. Also, only [2] is published at TOG in this July. In my opinion, they are more like concurrent works rather than previous works.

- [1] LayerDiff: Exploring Text-guided Multi-layered Composable Image Synthesis via Layer-Collaborative Diffusion Model
- [2] Transparent Image Layer Diffusion using Latent Transparency

In the rebuttal document and comments, the authors have provided the comparison results with LayerDiffusion qualitatively and quantitatively. I think that the additional experimental results provided by the authors support that the proposed method and LayerDiffusion have distinct properties and there are some scenarios when the proposed method works better (e.g., when we need more controllability by text, or layout image).

In terms of evaluation, I partially agree with Reviewer GbJP that this paper can include more baselines, but at the same time, I partially agree with the rebuttal, except LayerDiffusion, they chose the best methods that were available at the submission time. I personally recommend the authors to add more baseline methods despite they are not current state-of-the-art methods, to emphasize the contribution of the proposed method.

Except for the novelty and the evaluation issue, I think the remaining weaknesses are not significant issues.

Considering the advantage of the proposed method (high controllability, enabling more controllable factors compared to the existing state-of-the-arts -- including LayerDiffusion) (wSp1, S8j5, tSA3), and the novelty issue seems to be partially resolved by the rebuttal, I recommend acceptance of this paper.